# The Lithuanian Lung Cancer Screening Model: Results of a Pilot Study

**DOI:** 10.3390/cancers17121956

**Published:** 2025-06-12

**Authors:** Edvardas Danila, Leonid Krynke, Audronė Ciesiūnienė, Emilė Žučenkienė, Marius Kantautas, Birutė Gricienė, Dileta Valančienė, Ingrida Zeleckienė, Rasa Austrotienė, Gabrielė Tarutytė, Lina Vencevičienė

**Affiliations:** 1Clinic of Chest Diseases, Immunology and Allergology, Institute of Clinical Medicine, Faculty of Medicine, Vilnius University, Santariškių St. 2, 08661 Vilnius, Lithuania; 2Centre of Pulmonology and Allergology, Vilnius University Hospital Santaros Klinikos, Santariškių St. 2, 08661 Vilnius, Lithuania; 3Clinical Radiation Surveillance Division, Vilnius University Hospital Santaros Klinikos, Santariškių St. 2, 08661 Vilnius, Lithuania; leonid.krynke@santa.lt (L.K.); birute.griciene@santa.lt (B.G.); 4Centre for Informatics and Development, Vilnius University Hospital Santaros Klinikos, Santariškių St. 2, 08661 Vilnius, Lithuania; audrone.ciesiuniene@santa.lt (A.C.); gabriele.tarutyte@santa.lt (G.T.); 5Clinic of Internal Diseases and Family Medicine, Institute of Clinical Medicine, Faculty of Medicine, Vilnius University, Santariškių St. 2, 08661 Vilnius, Lithuania; emile.zucenkiene@santa.lt (E.Ž.); marius.kantautas@santa.lt (M.K.); lina.venceviciene@santa.lt (L.V.); 6Centre of Family Medicine, Vilnius University Hospital Santaros Klinikos, Santariškių St. 2, 08661 Vilnius, Lithuania; 7Department of Radiology, Nuclear Medicine and Medical Physics, Institute of Biomedical Sciences, Faculty of Medicine, Vilnius University, Santariškių St. 2, 08661 Vilnius, Lithuania; dileta.valanciene@santa.lt (D.V.); ingrida.zeleckiene@mf.vu.lt (I.Z.); 8Centre of Radiology and Nuclear Medicine, Vilnius University Hospital Santaros Klinikos, Santariškių St. 2, 08661 Vilnius, Lithuania; 9Project Management Department, Vilnius University Hospital Santaros Klinikos, Santariškių St. 2, 08661 Vilnius, Lithuania; rasa.austrotiene@santa.lt; 10Department of Research and Innovation, Faculty of Medicine, Vilnius University, M. K. Čiurlionio St. 21, 03101 Vilnius, Lithuania

**Keywords:** lung cancer, screening, pilot study

## Abstract

In the European Union, lung cancer causes roughly as many deaths as colorectal, breast, uterine, and ovarian cancers combined. Early lung cancer detection (screening) aims to identify cancer in its latent, clinically silent stage. Lung screening programs are a significant step forward in the fight for people’s lives against lung cancer. Unfortunately, their effectiveness remains limited due to low participation rates, with only a small proportion of the target population undergoing screening. By restricting screening to heavy smokers or former heavy smokers, traditional screening programs fail to detect a large portion of lung cancer cases. In Lithuania, a national lung cancer screening program was developed in 2024. Unlike traditional programs, the Lithuanian program screens participants regardless of their smoking history. A pilot study of the Program has been carried out and demonstrated that the Lithuanian lung cancer screening model is feasible, well-organized, and clinically valuable.

## 1. Introduction

Lung cancer is a major global health problem and the second most common cancer in both men and women [1]. However, among all malignancies, lung cancer is the leading cause of cancer-related deaths worldwide. In the European Union, lung cancer causes roughly as many deaths as colorectal, breast, uterine, and ovarian cancers combined [2]. This high mortality rate is due to several factors. First, lung cancer is often clinically “silent” when confined to the lung parenchyma, and it only causes non-specific symptoms (e.g., cough) when growing in the bronchi [3,4]. Additionally, even small, low-volume lung tumors tend to metastasize early, further complicating treatment and prognosis [5,6].

Early lung cancer detection (screening) aims to identify cancer in its latent, clinically silent stage [7]. If lung cancer is detected at an early stage, the 5-year survival rate can exceed 50%, compared to only ~3–5% in late-stage disease [8,9]. While the exact duration of the latent phase of lung cancer remains uncertain, existing data suggest it lasts about 5–6 years for non-small cell lung cancer [10,11]. However, due to the highly aggressive and rapid course of small-cell lung cancer, screening is likely not adequate for this subtype [12]. Many countries implement national, regional, or pilot lung cancer screening programs, demonstrating that early detection can reduce lung cancer mortality [13,14]. The cornerstone of screening is low-dose chest computed tomography (LDCT) [15]. LDCT is a safe screening method, as the emitted radiation dose (~1 mSv) is markedly lower than the annual background radiation and does not pose a substantial health risk [16].

Lung screening programs are a significant step forward in the fight for people’s lives against lung cancer. Unfortunately, their effectiveness remains limited due to low participation rates, with only a small proportion of the target population undergoing screening [17,18]. By restricting LDCT screening to heavy smokers or former heavy smokers, such programs fail to detect a large portion of lung cancer cases [19,20,21].

In Lithuania, a national lung cancer screening program (hereafter the Program) was prepared in 2024 [22]. The Lithuanian screening program aims to examine individuals between 50 and 70 years of age. This Program stands out from traditional screening programs in two key ways. First, eligibility is not limited by smoking status—participants are screened regardless of their smoking history. Second, the approach is comprehensive, maximizing the diagnostic potential of the LDCT method. In addition to detecting lung nodules and potential cancerous changes, CT scans will be actively assessed for significant incidental findings in the lungs and other organs covered by the scan [22]. We hypothesized that including never-smokers would uncover many otherwise missed pathologies.

Throughout the Program, people will be invited for screening every 3 years if no significant changes are detected on the initial LDCT. If significant findings are detected, further evaluation will depend on the specific findings. The Ministry of Health of the Republic of Lithuania tasked Vilnius University Hospital Santaros Klinikos (VUH SK), together with two other healthcare institutions (family medicine centers), to carry out a pilot study of this Program. This study aimed to evaluate the efficiency of the screening process and the Program’s potential efficacy [22]. The most essential process indicator was the participation rate of invited individuals. The key effectiveness indicators were the number of detected Lung-RADS 2022 (Lung CT Screening Reporting and Data System) category 4 nodules [23] and the number of significant incidental findings detected according to the 2023 guidelines for incidental findings in LDCT lung screening published by ERS (European Respiratory Society)/ESTS (European Society of Thoracic Surgeons)/ESTRO (European Society for Radiation Oncology)/ESR (European Society of Radiology)/ESTI (European Society of Thoracic Imaging)/EFOMP (European Federation of Organizations for Medical Physics) [24].

## 2. Methods

**Participant Recruitment.** Three family medicine centers were selected for the pilot Program: Family Medicine Center of VUH SK (Center 1), Vilnius Pašilaičiai Family Medicine Center (Center 2), and Elektrėnai Municipality Health Center (Center 3). Lists were compiled of individuals aged 50–70 years, enrolled at these centers, who did not have lung cancer (according to the current International Classification of Diseases, 10th Revision, Australian Modification ICD-10-AM, inclusion criteria).

Center 1 had about 12,000 enrolled participants during the pilot study, of whom 3924 met the target population criteria. Center 2 and Center 3 had about 5000 (902 targets) and 16,000 (3520 targets) participants, respectively. The lists of target group individuals from these centers were provided to the Coordinating Center. Two employees of the Coordinating Center contacted the individuals on the list in sequential order to inform them about the opportunity to participate in the Program. The call duration was not limited, but the length of each conversation was documented.

Individuals who expressed interest in the Program were asked two questions: (1) whether they had undergone chest or whole-body CT in the last 3 years, and (2) whether they were self-sufficient (not bedridden). The second question indirectly assessed the person’s Eastern Cooperative Oncology Group (ECOG) performance status, aiming to exclude those with ECOG ≥ 3 [25]. If the answer to the first question was “yes” or the person’s condition corresponded to ECOG ≥ 3 points, they were excluded from the Program.

If an individual met the Program’s inclusion criteria and agreed to participate, they were immediately scheduled for an LDCT at a designated date and time. Those who wished to take time to consider or consult their family doctor were allowed to do so. For those who agreed to participate, Coordinating Center staff also collected information about their smoking status.

To assess whether significant findings on LDCT were new, all pre-existing participants’ diagnoses were recorded into the Program database by ICD-10-AM code. The study was conducted with approval from Vilnius’s regional biomedical ethics committee, and all participants signed an informed consent form.

**Family Physician Team’s Role Before LDCT**. During the Coordinating Center’s phone call, the staff members encountered questions they could not answer, and they recorded in the Program database that the participant desired a remote consultation with their family physician team. Each business day, the family medicine center coordinator checked the database for participants requesting consultation and forwarded their details to the family physician team. A family doctor’s team member would then contact the participant, provide detailed information about the Program, and answer any further questions. At the end of the conversation, the team member would ask if the participant agreed to participate. They recorded the call duration in the database, and if the participant agreed, they scheduled the LDCT and noted the appointment date in the database. If the patient declined participation, the database recorded this as the reason for refusal.

**Chest LDCT Procedure.** All participants underwent LDCT at Center 1. CT exams were performed on a GE Revolution HD 64-slice CT scanner (2020) using a low-dose protocol. The CT protocol parameters were set so that participant radiation dose would meet international recommendations while image quality remained sufficient for diagnostic needs [26].

In developing the LDCT protocol, the recommendations, protocol guidelines, and technical standards of international societies were followed, including the American Association of Physicists in Medicine (AAPM) [27], the European Society of Thoracic Imaging (ESTI) [28], and the American College of Radiology/Society of Thoracic Radiology [29]. The medical physicists’ team aimed to ensure that all dose parameters remained below 1 mSv for the effective dose, 3 mGy for the CT dose index (CTDI), and 75 mGy·cm for the dose–length product (DLP) in a standard-sized participant. For participants weighing 50–80 kg, the target CTDI was approximately 0.8 mGy. The additional cancer risk from the CT scans was evaluated using the X-rayRisk.com calculator [30].

Low-dose chest CT examinations were conducted without intravenous or oral contrast material. Participants were scanned in the supine position with their arms raised above their heads, when feasible. All scans were performed during a single breath-hold at full inspiration to minimize motion artifacts. A standardized low-dose spiral protocol was utilized, covering the entire lung field from apices to bases. The tube voltage was set at 100 kVp, and automatic tube current modulation (Smart mA) was employed to adjust the tube current in real-time based on participant size and anatomical attenuation throughout the scan range. The tube rotation time was 0.4 s per rotation, the pitch factor was 1.38, the slice thickness was 0.625 mm, and the collimation width was 40 mm. Standard (body) and lung-specific reconstruction filters were used.

Five radiology technologists were involved in performing the LDCT examinations. The chest MDCT scans were evaluated by five radiologists with more than two years of clinical experience in interpreting chest CT examinations. Additionally, they were certified in a specialized lung cancer screening training course from the European Society of Thoracic Imaging (ESTI). They had completed the Vilnius University course on “Lung Cancer Screening and Early Diagnosis”.

All chest LDCT images were interpreted with the “Aview LCS Plus” artificial intelligence (AI) software package (Coreline Europe GmbH, Frankfurt a. Main, Germany). The AI software automatically detected lung nodules, measured nodule size and volume, and provided 3D visualization. The software enabled features, such as grouping nodules by Lung-RADS 2022 categories [23], semi-automatic editing of nodule boundaries, and assessment of additional findings like the coronary artery calcification score and presence of emphysema.

Lung findings were categorized according to the Lung-RADS v2022 reporting and data system [29]. Incidental findings were assessed according to the 2023 guidelines from ERS/ESTS/ESTRO/ESR/ESTI/EFOMP and related societies on managing incidental findings in LDCT lung cancer screening [24].

**Family Physician Team’s Role After LDCT**. The LDCT results for each participant were documented by a radiologist, who evaluated the findings, assigned any changes to the appropriate Lung-RADS category (0, 1, 2, 3, 4A, 4B, 4X), and entered the data into the Program database.

Based on the category of findings and any significant incidental findings, the family medicine center coordinator referred the patient for a follow-up consultation with the family doctor. Participants with category 0, 2, 3, 4A, 4B, or 4X findings were referred to the family physician in a priority order determined by the categories. If only category 1 findings were detected, the patient was referred to a nurse on the family doctor’s team for consultation.

A family physician or team member contacted the patient to explain the radiology report. Based on the category of LDCT findings, the family physician/team member informed the patient when a follow-up (surveillance) LDCT should be performed, after 3, 6, or 12 months (or sooner for category 0). If the changes were insignificant or no abnormalities were found, the participant was informed that they could participate in the Program again after 36 months. If significant incidental findings were detected, the patient was referred for a planned consultation with the appropriate physician of the relevant specialty for further evaluation and treatment. If a suspicious finding and/or significant incidental change was seen on LDCT, the patient was referred via a fast-track pathway to a pulmonologist or other specialist for prompt evaluation. In cases requiring a follow-up LDCT, the process and schedule for that exam were explained to the patient. When smokers in the study meet with their family physician to discuss findings related to pulmonary issues or other significant health concerns, they are advised to quit smoking.

After the family physician team member discussed the LDCT results with the patient by phone, the following parameters were recorded in the Program database:
Call duration: The length of the phone conversation.Incidental diagnoses: If significant comorbid diagnoses were identified, they were recorded by the ICD-10-AM code.Outcome of this stage: Recorded as one of the following—will be invited again by the Program, referred to a specialist physician, will no longer be asked by the Program, or a follow-up LDCT will be performed.

**Outcome Measures.** The pilot Program aimed to enroll 1000 individuals. We evaluated the following outcome measures of the pilot:
I.**Screening process (program implementation) metrics**


**Main indicators**


The number of people invited (i.e., contacted by phone).Number of people who underwent LDCT: how many attended the LDCT scan, and what percentage of all those invited does this represent, overall and stratified by sex (women/men) and smoking status (smokers/non-smokers).Number of people who agreed to LDCT but did not attend: the count of individuals who consented to LDCT but did not attend for the scan, and this number as a percentage of those who agreed.Invitation call duration: the average length of the phone invitation call, overall and for men, women, smokers, and non-smokers.Requests for physician consultation: how many invitees requested a family doctor consultation before deciding, as a number and percentage of all invited.Average LDCT appointment duration: the average time a screening visit took (from entering to exiting the CT suite).Average LDCT interpretation time: the average time for radiologists to interpret and report on baseline LDCT scan, with and without the aid of AI software.


**Other process indicators:**
Profile of those who refused participation: the breakdown of individuals who declined participation, categorized by sex and smoking status (each expressed as a percentage of all who refused).Reasons for declining participation in the pilot.
II.
**Screening effectiveness indicators:**

Lung nodules detected: the number of nodules detected, categorized by Lung-RADS v2022, with each category expressed as a percentage of all nodules.Significant incidental findings: the number of substantial incidental findings (stratified by ICD-10-AM diagnosis codes) and the percentage of newly identified (previously undiagnosed) findings.


**Study population.** Of 1920 individuals invited to participate in the Program, 459 (23.9%) refused, and 100 (9.0%) of all who agreed and registered did not attend the LDCT appointment. An additional 347 people had decided to participate but were not called for LDCT once the target sample size was reached. Thus, 1014 individuals ultimately underwent LDCT screening between 26 September 2024 and 14 February 2025, which is slightly above the target sample, as some already-scheduled individuals were scanned even after the 1000 target was reached. The demographics of the study population are shown in Table 1.

## 3. Results

**Invitation process efficiency.** The average phone call inviting an individual to the Program lasted just over 4 min (Table 2). Therefore, one Coordinating Center staff member could comfortably make about 50 invitation calls in an 8 h workday, i.e., roughly 1000–1150 calls per month (on business days). Only six individuals (0.5% of those invited) wanted to consult their family doctor before participating.

The average time to interpret and report on a baseline LDCT scan was 8 min using the AI software, compared to 10 min without AI assistance. Using AI, a radiologist could comfortably evaluate and report ~45 LDCT scans in one 7.5 h workday (~900–1000 per month), whereas without AI, a radiologist could assess and report ~40 scans per day (~800–900 per month).

For follow-up (surveillance) LDCT scans, the interpretation time is expected to be slightly longer: about 10 min with AI (an increase of ~2 min, or 25% longer than the baseline scan) and about 13 min without AI (an increase of ~3 min, or 30% longer). The results of the study organization process are presented in Table 2.

**Screening effectiveness outcomes.** A total of 996 lung nodules were detected on the LDCT scans. Of them, 267 (26.3% of all participants) nodules in categories 2–4 by Lung-RADS v2022 were detected. Fourteen participants (1.4% of those screened) had nodules classified as Lung-RADS category 4 (suspicious, very suspicious, highly suspicious). In addition, one participant with a Lung-RADS category 2 nodule was confirmed to have squamous cell carcinoma arising from peripheral lung changes; these changes did not meet Lung-RADS 2022 criteria for a nodule but were classified as significant incidental. The distribution of nodules in categories 2–4 by Lung-RADS v2022 is shown in Table 3.

**Incidental findings**: In total, 305 participants (30.1% of those screened) had at least one significant incidental finding on LDCT. Among these, 263 (86.2%) were newly diagnosed abnormalities (i.e., previously undetected). The significant incidental findings included cardiovascular (208 cases; 20.5% of all participants), pulmonary or mediastinal (72 cases; 7.1%), and other findings (25 cases; 2.5%).

The most frequently identified significant incidental findings were coronary artery calcification—178 cases (58.4% of all significant incidental findings), and aortic valve calcification—20 cases (6.6%).

Of the 72 significant pulmonary and mediastinal incidental findings, 52 (72.2%) were newly diagnosed. The most common significant incidental findings are presented in Table 4, while newly diagnosed morbidities (based on new significant incidental abnormalities) are detailed in Table 5 and Appendix A.

After LDCT and the family physician’s evaluation, participants were referred for further care according to the Program protocol. Depending on the LDCT findings and the consultation, participants were directed either to a pulmonologist via the fast-track or to a scheduled (routine) consultation with a pulmonologist or other physician. Table 6 summarizes the referrals for specialist consultations. The discussion between the family physician and patient about the LDCT results (as described in Methods) lasted about 3 min on average.

**Radiation exposure:** Radiation dose data for the LDCT examinations are presented in Table 7.

## 4. Discussion

The most important findings of this pilot study are as follows: (1) 76.1% of the target population invited agreed to participate in screening, (2) Lung-RADS v2022 category 4 nodules were found in 1.4% of participants, and (3) 25.9% of participants had at least one newly identified significant incidental finding. Below, we discuss these results in the context of existing evidence and the unique aspects of the Lithuanian lung cancer screening model.

Low participation of the target population in lung cancer screening programs has undermined their impact and feasibility. For example, only up to 6% of eligible individuals undergo screening in the United States [18], although more recent data showed a national average participation of around 16% [31]. Participation is around 19% in Canada, 6–31% in China, ~38% in South Korea, and ~50% in Japan [32]. In the Yorkshire Lung Screening Trial in the UK, only 29.1% of those invited attended and underwent LDCT screening [33]. Only in Estonia, a neighbor in the Baltic region, comparable results were reported, with 79.3% of people at high risk of lung cancer participating in a pilot study [34]. However, compared to our pilot study, the Estonian study required more time and resources to identify individuals’ eligibility for screening requirements, inviting individuals, and involving medical staff.

The participation rate in the study far exceeded our expectations. Before the study, we planned for around 30% of those invited to participate. Our estimates were based on the experience of other countries and the rate of participation in other screening programs in Lithuania. In contrast, this study achieved a remarkably high participation rate of 76.1% of those invited.

The high participation rate observed in this cohort is not entirely explainable; however, several key factors likely contributed to this, including (a) personal telephone invitation—a live, one-on-one conversation, as opposed to impersonal mailings, and (b) the comprehensive nature of the Program, which emphasized that a single, non-invasive test could detect multiple potential diseases beyond lung cancer. Additionally, a short-term, moderate-intensity informational campaign was conducted in national and commercial media, highlighting the importance of the Program and announcing that the pilot study likely fostered a positive attitude toward the Program among potential participants. In our research, calls were made by Coordinating Center staff. It is likely that if the calls had come from a member of the person’s own family doctor’s team (rather than an unfamiliar “central” staff member), participation would have been even higher—some who said they were not interested or feared radiation or results might have agreed if approached by their known healthcare provider. The participation rate might have been even higher if it hadn’t been for the Christmas and New Year period, as well as the short duration of the study, which limited the availability of LDCT times for participants.

One hundred individuals (9%) who consented to participate and were scheduled for a chest LDCT did not attend the examination. It is important to note that the pilot screening study was conducted during the cold season (a period of increased respiratory infections), making it highly likely that some individuals who had initially agreed to participate did not attend due to an acute viral respiratory infection. According to the Program protocol, chest LDCT must be performed no earlier than three months after resolving an acute respiratory infection.

We believe the cost associated with an individualized telephone invitation (which in our study took, on average, up to 5 min) is incomparably lower, and the benefit is far greater than the cost of treating advanced lung cancer [35,36]. One dedicated employee can comfortably call and speak with about 50 target individuals during an 8 h workday, or ~1000 individuals per month. Several studies have shown that telephone-based smoking cessation interventions are cost-effective in the lung screening setting [37,38]. Moreover, telephone-based assessments of frailty and eligibility can help identify which individuals are suitable for lung cancer screening [39,40] and for navigating high-risk or vulnerable individuals to screening [41]. Indeed, a phone call is more effective than a mailed invitation in recruiting participants [33].

The Lithuanian Lung Cancer Screening program aims to include non-smokers and smokers. This decision was based on several considerations. First, lung cancer is increasingly occurring in never-smokers, and this trend is expected to continue [21]. Importantly, the effectiveness of LDCT screening is not dependent on the smoking status of those screened [42]. Second, by screening only heavy smokers or ex-smokers, 50–70% of lung cancer cases go undetected [19,20]. Third, it would be socially unjust to only screen smokers (who are personally responsible for a known risk factor) for a potentially deadly disease while ignoring and not screening non-smokers, including those who are exposed to second-hand smoke. In summary, excluding never-smokers from screening would miss many cases and raise ethical equity issues.

Of the fourteen individuals with Lung-RADS category 4 nodules in our pilot, six were smokers, and eight were non-smokers. As noted earlier, roughly 16–27% of category 4 nodules are malignant, with variability by subcategory. Specifically, about 5–15% of 4A nodules, 15–36% of 4B nodules, and 6–77% of 4X nodules are ultimately diagnosed as cancer [43,44,45,46].

Due to the limited sample size and short duration of this pilot study, the “number needed to screen” to detect a single case of lung cancer and participants’ survival outcomes were not determined. However, this figure will be calculated and reported once follow-up scans and additional diagnostic evaluations are completed. Nonetheless, based on current findings, the proportion of participants with Lung-RADS 4 findings (approximately 1.4%) is consistent with the prevalence of such findings in the general population [23]. This was expected, given that our study included both smokers and non-smokers. The distribution of participants by smoking status aligns with national smoking prevalence statistics in Lithuania. According to the latest data from the State Data Agency, the proportion of non-smokers in the Lithuanian population was 76.3%, while the proportion of smokers was 23.7% [47].

Further modeling, calculation, and analysis of the costs and effectiveness of the Program will be conducted, considering not only the detection of lung cancer but also the identification of significant comorbidities.

Previous studies have found that participation in lung cancer screening yields little or no improvement in overall survival for smokers [15,48,49,50,51]. In contrast, for never-smokers, the benefit of lung cancer screening is most significant and most evident. Notably, because smokers face high competing mortality from other causes, some studies suggest that screening them yields smaller overall survival benefits than screening never-smokers, who are otherwise at lower risk of death [19,52]. This evidence further supports the inclusion of non-smokers in screening, as planned in Lithuania, to maximize the Program’s life-saving potential.

The value of LDCT as a comprehensive screening tool is well-supported by evidence [24]. With increasing life expectancy and an aging population, the prevalence of comorbidities is rising. In our study, significant incidental findings were actively assessed. Newly diagnosed significant pulmonary and mediastinal findings were identified in 5.1% of participants, cardiovascular findings in 18.7%, and other significant findings in 2.1%. In total, 17.6% of all participants were referred to a medical specialist (pulmonologist, cardiologist, or other) after their family physician evaluated the newly detected findings in the lungs or other organs as clinically relevant. Following chest LDCT, pulmonary tuberculosis was suspected in two asymptomatic individuals. After further diagnostic testing, tuberculosis was confirmed in one patient. The second patient has not yet returned for additional evaluation.

There is no doubt that the importance of such comprehensive screening will only increase over time [53]. A recent study in Lithuania [54,55] showed that many people over 50 have multiple significant chronic conditions, which LDCT could detect before clinical manifestation [54,55]. In our study, the percentage of participants with significant incidental findings (30.1%) was higher than the ~13–19% reported in other studies [56,57]. This is likely because the evaluation considered the radiological significance of the findings and their clinical significance in determining further patient care. Comprehensive screening inevitably detects many incidental abnormalities. At the same time, our Program managed these findings systematically; we recognize that not all incidental findings will be clinically significant, and their discovery can sometimes lead to unnecessary follow-up procedures. However, our protocol attempted to mitigate this by focusing on significant incidental findings requiring specialist referral.

If implemented, pilot study results from the Lithuanian national health system perspective indicate that the Program will detect diseases before they become clinically apparent in many individuals through a single, non-invasive, periodic test. The changes detected (early disease) strongly incentivize individuals to change harmful habits and adopt preventive measures. In this way, at least part of the progression of the disease can be prevented.

If the Program is not implemented, most diseases will progress and become clinically evident later. They will still be diagnosed, but this will occur later. A greater proportion of these diseases will be more severe, potentially with complications. Diagnosing them will take longer and be more expensive, as many cases require differential diagnosis or specific diagnostic steps. For some diseases, valuable time will be lost in preventing progression, complications, and related events.

Three main factors were considered when determining the screening interval (one scan every three years) for the Lithuanian Program. First was the interval that would be sufficiently safe, given the natural history of non-small cell lung cancer. Second was the anticipated compliance of the target population with repeat LDCT scans. Third was the accessibility of the service to the target population (resources and logistics of providing scans).

A clearly “safe” interval for LDCT screening has not been definitively established and varies across studies [58]. Modeling studies suggest 1–3 years interval for high-risk individuals and 5–10 years interval for those at lower risk [59,60,61]. Several studies have shown poor participant adherence to annual repeat LDCT screening [62,63,64]. We believe that a maximum of seven scans (one every 3 years between ages 50 and 70) will be far more acceptable to participants than twenty scans (annual screening over that period). We expect participation and overall program effectiveness to improve with our 3-year interval model. Moreover, such a screening frequency will avoid unnecessary LDCT rounds and reduce cumulative radiation exposure and costs [16]. Further research and ongoing monitoring will ultimately determine the success of this approach. Regardless, current lung cancer screening models remain suboptimal and warrant improvement.

The Program’s accessibility to the target population is crucial. When the Lithuanian lung cancer screening program was developed, the country had a population of 2.9 million people [65]. The entire target population was approximately 0.78 million individuals. The target population for LDCT screening was about 260,000 for one year. At that time, 56 CT scanners in 35 medical institutions met the Program’s quality requirements (this number is expected to increase soon). The distribution of CT scanners in Lithuania is presented in Figure 1. The longest distance from the most remote residential area to a CT scanner meeting the Program’s requirements is 89 km, an average distance of 50 km.

On average, each CT scanner would need to perform an additional 4600 chest LDCT scans per year if 100% of the target population participated. If approximately 76% of the target population participated, as observed during the Program’s pilot phase, a total of around 198,000 chest LDCT scans would be required annually, corresponding to approximately 3500 additional scans per CT scanner per year. Our pilot study showed that even in a hectic university hospital, one CT scanner can accommodate an additional 1000 scans in approximately four months without disrupting daily clinical operations. Since CT scanners in most minor and peripheral healthcare facilities are used significantly less, especially on weekends, service accessibility should be adequate nationwide.

Ultra-low-dose and ultra-fast chest CT protocols and AI-assisted programs are becoming part of daily clinical practice [66,67,68]. A previous study [69] found that the average cumulative effective dose after ten years of annual screening was 9.3 mSv for men and 13.0 mSv for women. Depending on the participant’s age and gender, the risk of developing lung or other cancers due to radiation exposure ranged from 1.4 to 8.1 cases per 10,000 screened individuals. According to the Biologic Effects of Ionizing Radiation VII assessment, among 10,000 participants undergoing ten annual LDCT scans, the radiation exposure itself could result in 4.6 cancer cases.

In this pilot study, the effective dose for a standard-sized participant and a participant group weighing 50 kg to 80 kg was estimated at approximately 0.5 mSv (the typical diagnostic chest CT effective dose is ~6 mSv). If LDCT is performed once every three years, the projected radiation exposure for a standard-sized participant would be 3.5 mSv (seven scans over the target population’s screening period). The estimated additional cancer risk due to radiation exposure would be lower—1.8 cases in men (aged 50) and 2.4 cases in women (aged 50) per 10,000 screened participants.

As the national screening program is rolled out, recommended parameters for low-dose CT protocols are specified for all participating centers (CTDI < 3 mGy, DLP < 75 mGy·cm, and effective dose < 1 mSv for standard-sized participants). Adherence to these dose constraints is expected and will be closely monitored. Importantly, the dose levels achieved in the pilot study were substantially lower than these recommended thresholds, demonstrating that even lower-dose protocols can be effectively used without compromising image quality. This experience is expected to encourage other medical sites to further optimize and adopt even lower dose protocols whenever feasible.

To ensure consistency and quality, the Program requires clinical audits, retrospective dose report analysis, and external quality assessments by expert medical physicists and radiologists. The Lithuanian Ministry of Health will oversee the process and implement corrective actions if necessary. We believe this robust audit and quality assurance strategy will help maintain low radiation doses and high diagnostic quality, ensuring that the benefits observed in the academic center can be replicated nationwide.

Undoubtedly, as CT technology rapidly advances, implementing filtration, dose reduction techniques, AI, specialized detectors, new X-ray generation methods, and improved algorithms will further reduce participants’ exposure to ionizing radiation.

With technological advancements, LDCT should increasingly replace conventional chest radiography as a preventive screening tool. Its use should be expanded comprehensively, integrating the detection of lung cancer with the diagnosis of other diseases to maximize its clinical utility [70,71,72,73]. AI-assisted programs detect and assess lung nodules (the primary goal of the lung cancer screening program) and identify lung emphysema, interstitial changes, bronchiectasis, coronary artery conditions, and other findings. These tools enhance the efficiency of radiologists’ work, ultimately improving patient outcomes [68,74,75,76].

This study has strengths and limitations. One key strength is that it prospectively tested the national lung cancer screening concept’s effectiveness and process feasibility. The study was conducted in real-life conditions without disrupting routine hospital clinical practice. Throughout the pilot study, representatives from Lithuania’s medical authorities (the Ministry of Health and the National Health Insurance Fund) monitored the process in real time. A data collection tool and a real-time data visualization dashboard were developed using the Power BI platform, which displayed all processes and results related to the pilot study (see Appendix A). This pilot study was successfully executed in real-life conditions and demonstrated the feasibility and potential effectiveness of the model. It can be used for lung cancer screening at both the institutional and national levels.

The study also has a few limitations. It was conducted over a limited timeframe and included a relatively small sample size of 1014 participants, which may impact the generalizability of the findings. It should be noted that the study was conducted by experienced and motivated university hospital staff. Radiologists used an AI-assisted program, which increased efficiency, automated measurements, and the detection of significant changes, reducing the image evaluation and reporting time to an average of eight minutes. Medical physicists developed low-dose CT protocols for participant examinations. The results of this study may differ in other settings.

## 5. Conclusions

This pilot lung cancer screening study demonstrated that the Lithuanian lung cancer screening model is clinically valuable and well-organized in process management. The model is clinically significant, as the number of Lung-RADS v2022 category 4 nodules detected was comparable among smokers and non-smokers. Furthermore, many previously undiagnosed significant comorbidities, such as coronary heart disease and chronic obstructive pulmonary disease, were revealed. The high participation rate among the target population and efficient organizational management indicate that this screening model can be effectively implemented in Lithuania. Moreover, we believe the model could be successfully implemented in different countries.

## Figures and Tables

**Figure 1 cancers-17-01956-f001:**
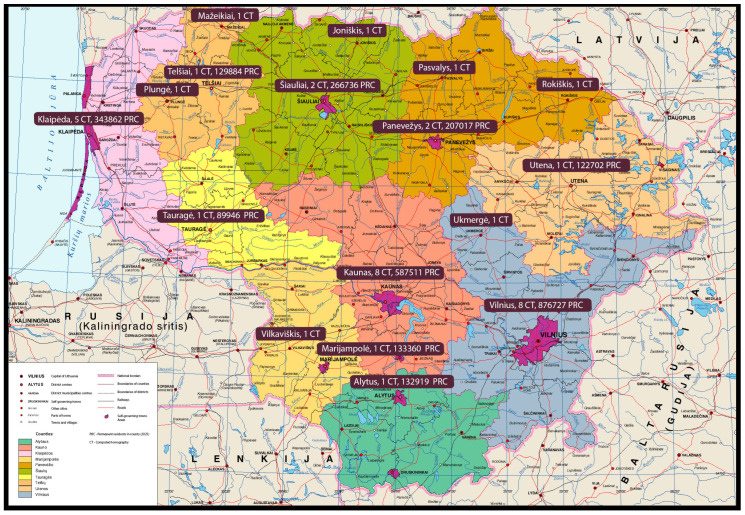
Projected distribution of CT scanners meeting technical requirements for the Lithuanian lung cancer screening program. Boxes indicate the city names and the estimated number of available CT scanners.

**Table 1 cancers-17-01956-t001:** Demographics of the study population.

Metric/Parameter	Value
Number of participants (LDCT performed)	1014
Sex—male/female (%)	528 (52.1%)/486 (47.9%)
Age, years (mean ± SD)	60.5 ± 5.2
Smoking status—yes/no/unknown (%)	217 (21.4%)/795 (78.4%)/2 (0.2%)
Smoking history (current smokers):	
Traditional tobacco (pack—years, mean ± SD)	23.63 ± 16.50
E-cigarettes (years, mean ± SD)	4.5 ± 4.2

**Table 2 cancers-17-01956-t002:** Key program implementation results (screening process outcomes).

Metric/Parameter	Value
**Invitation call duration,** minutes (mean)	
Men/Women	4.1/4.4
Non-smokers/Smokers	4.1/4.7
**LDCT appointment duration**	
(check-in to check-out), minutes (mean)	15
**LDCT interpretation time,** minutes (mean)	
With AI assistance	8
Without AI assistance	10
**Refusals**	
Number (% of invited)	459 (23.9%)
Men/Women *	216 (47.1%)/243 (52.9%)
Non-smokers/Smokers/Unknown **	8 (1.7%)/4 (0.9%)/447 (97.4%)
**Reasons for refusal**	
“Not interested”	234 (50.9%)
“Other” ***	189 (41.2%)
“Fear of radiation”	9 (1.9%)
“Fear of possible results”	5 (1.1%)
“No reason given”	22 (4.8%)

AI—artificial intelligence. * Gender distribution does not statistically significantly differ from participants who agreed to participate (0.07, Pearson’s Chi-squared test). ** Smoking status distribution does not statistically significantly differ from participants who agreed to participate (0.306, Fisher’s Exact Test). *** The most common “Other” reasons for refusal were inconvenient timing; inability to talk during work (leading to immediate refusal); shift work schedule making it hard to plan an appointment; having caught a cold and not being able to come; belief that the service was not relevant because the person did not belong to a risk group; and thinking the call might be a scam due to not having heard about the Program.

**Table 3 cancers-17-01956-t003:** Distribution of detected lung nodules by Lung-RADS v2022 category.

Category (Description)	Number (% of all 1014 participants)
**2 (Benign)**	**222 (21.9%)**
Non-smokers	188 (84.7%)
Smokers	34 (15.3%)
**3 (Probably benign)**	**31 (3.1%)**
Non-smokers	21 (67.7%)
Smokers	10 (32.3%)
**4A (Suspicious)**	**12 (1.2%)**
Non-smokers	8 (66.7%)
Smokers	4 (33.3%)
**4B (Very Suspicious)**	**1 (0.1%)**
Non-smokers	0
Smokers	1 (100%)
**4X (Highly Suspicious)**	**1 (0.1%)**
Non-smokers	0
Smokers	1 (100%)
**In total**	**267 (26.3%)**
Non-smokers	217 (81.3%)
Smokers	50 (18.7%)

Fourteen participants had Lung-RADS 4 nodules (as noted above), and one additional cancer (squamous cell carcinoma) was diagnosed in a participant with a category 2 nodule. Distribution of smokers and non-smokers in Lung-RADS 4 does not differ statistically significantly (0.483, Fisher’s Exact Test).

**Table 4 cancers-17-01956-t004:** Most frequently identified significant incidental findings.

Significant Incidental Finding	Number (% of all participants)
**In the Lungs and Mediastinum**	
Consolidation	8 (0.8%)
Interstitial lung changes	15 (1.5%)
Emphysema	16 (1.6%)
Bronchiectasis	14 (1.4%)
Suspected active tuberculosis	2 (0.2%)
Pleural changes	2 (0.2%)
Diaphragmatic changes	6 (0.6%)
Mediastinal mass	5 (0.5%)
Mediastinal lymphadenopathy	2 (0.2%)
Other	2 (0.2%)
**In Other Organs**	
Coronary artery calcification	178 * (17.6%)
Aortic valve calcification	20 (2.0%)
Aortic dilation	9 (0.9%)
Thyroid nodules	4 (0.4%)
Pericardial effusion	1 (0.1%)
Esophageal changes	1 (0.1%)
Breast lesions	1 (0.1%)
Lesions in parenchymal abdominal organs	9 (0.9%)
Bone changes	5 (0.5%)
Other	5 (0.5%)
**Total**	**305 (30.1%)**

* A total of 116 (65.2%) non-smokers and 60 (33.7%) smokers (2 participants did not respond). Among the group with no coronary artery calcification (*n* = 836), there were 679 (81.2%) non-smokers and 157 (18.8%) smokers. More smokers were in the coronary artery calcification group (*p*-value < 0.001, Pearson’s Chi-squared test).

**Table 5 cancers-17-01956-t005:** Participants with at least one newly identified significant incidental finding.

New Significant Incidental Finding	Count (% of All Participants)
One finding *	225 (22.2%)
Two findings **	32 (3.2%)
Three or more findings	4 (0.4%)

* A total of 154 (68.75%) non-smokers and 70 (31.25%) smokers (1 participant did not respond). Statistically significantly more smokers were in the one finding group, compared to the participants without findings (*p*-value < 0.001, Pearson’s Chi-squared test). ** Nineteen (59.4%) non-smokers and thirteen (40.6%) smokers. Statistically significantly more smokers were in the two findings group, compared to the participants without findings (*p*-value 0.002, Pearson’s Chi-squared test).

**Table 6 cancers-17-01956-t006:** Referrals for specialist consultations based on LDCT.

Referred to	Number of Participants (% of All Screened)
Pulmonologist (fast-track)	3 (0.3%)
Pulmonologist	128 (12.6%)
Cardiologist	27 (2.7%)
Abdominal surgeon	5 (0.5%)
Urologist	1 (0.1%)
Endocrinologist	2 (0.2%)
Other specialist	12 (1.2%)
Total referred to any specialist	178 (17.6%)

**Table 7 cancers-17-01956-t007:** Participant radiation dose metrics for chest LDCT scans.

Parameter	Value, Median (Min; Max)
All participants	
Weight (kg)	82 (47; 158)
Height (cm)	171 (150; 198)
Standard-sized participant *	
CTDI (mGy)	0.8 (0.6; 1.6)
DLP (mGy·cm)	33.6 (23.8; 63.7)
Effective dose (mSv)	0.5 (0.3; 0.9)
Participants 50–80 kg	
CTDI (mGy)	0.8 (0.5; 2.6)
DLP (mGy·cm)	34.0 (20.3; 92.9)
Effective dose (mSv)	0.5 (0.3; 1.3)

* Standard-sized participant defined as 70 ± 5 kg, 170 ± 5 cm. CTDI: CT dose index. DLP: dose–length product. The effective dose was calculated as DLP × 0.014.

## Data Availability

The data presented in this study are available on reasonable request from the corresponding author.

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
