# Peer review of "The Lithuanian Lung Cancer Screening Model: Results of a Pilot Study"

_cancers, 2025, doi:10.3390/cancers17121956_

Round 1
Reviewer 1 Report
Comments and Suggestions for Authors
Did you conduct the study or the data is provided by the MoH of your country?
I suggest to include the statistical analysis on the data provided in the present study such as ANOVA, post-hoc analysis, regression, PCA, factor analysis etc.
Others are fine.
Author Response
Thank you very much for your careful review and constructive comments.
R1: Did you conduct the study or the data is provided by the MoH of your country?
RESPONSE: We, the authors of the manuscript, carried out the study. MoH was the ordering and supervising authority for the study.
R1: I suggest to include the statistical analysis on the data provided in the present study such as ANOVA, post-hoc analysis, regression, PCA, factor analysis etc.
RESPONSE: Thank you very much for your suggestion. As you recommended, we analyzed the data statistically and added it to the manuscript.
R1: Others are fine.
RESPONSE: Thank you very much for your positive comment.
Reviewer 2 Report
Comments and Suggestions for Authors
The manuscript covers a pilot study of a lung cancer screening programme for Lithuania. This is timely and of interest to many. The pilot is pragmatic in terms of set up (1000 participants screened; only one CT scanner in use) and appears to have achieved its objective demonstrating feasibility. Of particular interest is that the screening is not restricted to a risk group (typically smokers) but includes everyone of a certain age. The manuscript should specify what the current prevalence of smoking is in Lithuania to determine if there is a bias in the number of people actually agreeing to participate. If more smokers than non-smokers refuse to participate there would not only be bias but the screening may possibly also miss an important target group. The numbers in table 1 even suggest the opposite with a high proportion of smokers included (unless I underestimate the prevalence of smoking in the population).
The dose levels aimed for and achieved are impressive. Are there any audits planned to ensure what happens at a state of the art academic centre can be achieved everywhere?
A few more details on the scanning procedure (and the standardisation thereof) would be useful. Slice thickness? Breath hold?
The detailed description of incidental finding is valuable but I would suggest to put large data such as table 5 in supplementary materials.
On the other hand it would be interesting to see some actual cost estimates. I assume a detailed health economics investigation was conducted and a few numbers would be of interest even if a separate publication may be planned.
I wonder also if the authors could make some comments about plans (or otherwise) to combine screening with smoking cessation counselling as is done in other screening programmes.
Finally, the manuscript is well written and structured and the discussion adds value. I would consider participants not be called patients throughout the manuscript (eg line 197)
Specific comments:
The abstract could mention features of the planned screening programme, eg the three year cycle
The statement commencing in line 386 sounds too speculative to me: there is a lot of complexity such as participant (self) selection and stage migration that make this hard to prove.
Author Response
We sincerely appreciate all insightful comments, corrections, and suggestions that helped us to improve the quality of our manuscript and carry out the revision of the paper.
R2: The manuscript covers a pilot study of a lung cancer screening programme for Lithuania. This is timely and of interest to many. The pilot is pragmatic in terms of set up (1000 participants screened; only one CT scanner in use) and appears to have achieved its objective demonstrating feasibility. Of particular interest is that the screening is not restricted to a risk group (typically smokers) but includes everyone of a certain age.
RESPONSE: Thank you very much for your positive comment.
R2: The manuscript should specify what the current prevalence of smoking is in Lithuania to determine if there is a bias in the number of people actually agreeing to participate. If more smokers than non-smokers refuse to participate there would not only be bias but the screening may possibly also miss an important target group. The numbers in table 1 even suggest the opposite with a high proportion of smokers included (unless I underestimate the prevalence of smoking in the population).
RESPONSE: You raise a very valid point. Thank you very much. According to the latest data from the State Data Agency, the proportion of non-smokers in the Lithuanian population was 76.3%, while the proportion of smokers was 23.7%. Considering the numbers of smokers 21.4% and non-smokers 78.4% who participated in the screening program, it is clear that the distribution of participants by smoking status aligns with national smoking prevalence statistics in Lithuania.
Among those who declined to participate in the screening program, 8 individuals (1.74%) were from the non-smoker group, and 4 individuals (0.9%) were from the smoker group. However, for the majority of those who declined—447 individuals (97.4%)—the smoking status is unknown, which means its potential impact on the study sample cannot be evaluated.
The smoking status of participants in the screening program matched the smoking prevalence reported by the State Data Agency in Lithuania. Therefore, it can be concluded that there was no selection bias in recruiting the study sample. We have included the data on the current smoking prevalence in Lithuania in the manuscript.
We have identified two smoking-related proofreading errors in the original version of the manuscript in Tables 1 and 2, which we deeply regret. The revised version contains the corrected information.
R2: The dose levels aimed for and achieved are impressive. Are there any audits planned to ensure what happens at a state of the art academic centre can be achieved everywhere?
As the national screening program is rolled out, recommended parameters for low-dose CT protocols are specified for all participating centers (CTDI < 3 mGy, DLP < 75 mGy·cm, and effective dose < 1 mSv for standard-sized patients). Adherence to these dose constraints is expected and will be closely monitored. Importantly, the dose levels achieved in the pilot study were substantially lower than these recommended thresholds, demonstrating that even lower-dose protocols can be effectively used without compromising image quality. This experience is expected to encourage other medical sites to further optimize and adopt even lower-dose protocols whenever feasible.
To ensure consistency and quality, the Program requires clinical audits, retrospective dose report analysis, and external quality assessments by expert medical physicists and radiologists. The Lithuanian Ministry of Health will oversee the process and implement corrective actions if necessary. We believe this robust audit and quality assurance strategy will help maintain low radiation doses and high diagnostic quality, ensuring that the benefits observed in the academic center can be replicated nationwide.
We have added a comment about this potential challenge to the manuscript.
R2: A few more details on the scanning procedure (and the standardisation thereof) would be useful. Slice thickness? Breath hold?
RESPONSE: Thank you for this point. We have added a short information about the scanning procedure, including parameters like slice thickness and breath-hold instructions, to the manuscript.
R2: The detailed description of incidental finding is valuable but I would suggest to put large data such as table 5 in supplementary materials.
RESPONSE: Thank you very much for your suggestion. We have moved Table 5 to the supplementary materials.
R2: On the other hand it would be interesting to see some actual cost estimates. I assume a detailed health economics investigation was conducted and a few numbers would be of interest even if a separate publication may be planned.
RESPONSE: You raise a very valid point. Thank you very much. We are now developing a costing model and calculating the full costs of such a Programme in various aspects. Yes, a separate publication is planned. At this stage, we can say that the personnel resources cost of this project was approximately 18,000 EUR.
R2: I wonder also if the authors could make some comments about plans (or otherwise) to combine screening with smoking cessation counselling as is done in other screening programmes.
RESPONSE: Thank you for your valuable question. At this stage, we do not plan to align these programmes. There are already active anti-tobacco activities in Lithuania. The essence of our programme is to make the pathway to LDCT as short and quick as possible for individuals. However, in cases where smokers in this programme present to their GP for a discussion of the findings due to pulmonary findings or significant concomitant findings, they will be advised to quit smoking.
R2: Finally, the manuscript is well written and structured and the discussion adds value. I would consider participants not be called patients throughout the manuscript (eg line 197)
RESPONSE: Thank you for this remark. We fully agree and have corrected the manuscript accordingly.
R2: The abstract could mention features of the planned screening programme, eg the three year cycle.
RESPONSE: Thank you very much for your suggestion. We added this information to the abstract.
R2: The statement commencing in line 386 sounds too speculative to me: there is a lot of complexity such as participant (self) selection and stage migration that make this hard to prove.
RESPONSE: Thank you for this remark. We removed this statement from the manuscript.
Reviewer 3 Report
Comments and Suggestions for Authors
The manuscript is suitable for the journal scope. I suggested minor revision after the author has considered some questions below
- In the introduction, please write combine the paragraph to make it more comprehensive. For example: lines 72-82; lines 98-100....
- In conclusion, please write more details 1-2 sentences to conclude, why this model is clinically valuable and well-organized in process management?
- If the patient detects their lung cancer early at the first stage, Does the Lithuanian lung cancer screening model provide clinically valuable outcomes? What stages does the author focus the best?
- What kind of lung cancer does the author focus on? For example: large cell lung cancer H460, Lung cancer A549, mucoepidermoid H292…? Why does the author select this type of cancer? Please write in the manuscript.
- How well does LDCT detect early-stage lung cancer in individuals aged 50 to 70, regardless of smoking history?
- What is the prevalence of clinically significant signals, including lung cancer and other diseases, in individuals undergoing low-dose CT (LDCT) lung cancer screening regardless of smoking status?
- Based on the study “The Lithuanian Lung Cancer Screening Model: Results of a Pilot Study,” can it be concluded whether males or females smoke tobacco more frequently?
Author Response
We are grateful for the time and energy you expended on our behalf.
R3: The manuscript is suitable for the journal scope. I suggested minor revision after the author has considered some questions below
RESPONSE: Thank you very much for your positive comment.
R3: In the introduction, please write combine the paragraph to make it more comprehensive. For example: lines 72-82; lines 98-100....
RESPONSE: Thank you very much for your suggestion. We have made the appropriate changes to the manuscript.
R3: In conclusion, please write more details 1-2 sentences to conclude, why this model is clinically valuable and well-organized in process management?
RESPONSE: Thank you very much for your suggestion. We have made the appropriate changes to the manuscript.
R3: If the patient detects their lung cancer early at the first stage, Does the Lithuanian lung cancer screening model provide clinically valuable outcomes? What stages does the author focus the best?
RESPONSE: Thank you for this remark. The programme's key long-term goal for lung cancer is to detect early stages. We understand that in the first few years, there will be a wide range of lung cancers detected, with more late-stage cases. However, if the programme works well, the majority of lung cancers detected later on will be early stage.
R3: What kind of lung cancer does the author focus on? For example: large cell lung cancer H460, Lung cancer A549, mucoepidermoid H292…? Why does the author select this type of cancer? Please write in the manuscript.
RESPONSE: Thank you for this point. The programme aims to detect lung cancer in any type. However, studies by other authors have shown that such a programmes are probably not effective in detecting small cell lung cancer at an early stage.
R3: How well does LDCT detect early-stage lung cancer in individuals aged 50 to 70, regardless of smoking history?
RESPONSE: Thank you for this remark. By examining non-smokers, we aim to capture the maximum possible number of lung cancer cases, given that at least 30 percent of people with lung cancer are non-smokers.
R3: What is the prevalence of clinically significant signals, including lung cancer and other diseases, in individuals undergoing low-dose CT (LDCT) lung cancer screening regardless of smoking status?
RESPONSE: You raise a very valid point. Thank you very much. We have added information to the manuscript about the significant findings, depending on the participants' smoking status.
R3: Based on the study “The Lithuanian Lung Cancer Screening Model: Results of a Pilot Study,” can it be concluded whether males or females smoke tobacco more frequently?
RESPONSE: Thank you very much for this question. Unfortunately, no. The data from this study does not give us a reasonable picture of the distribution of smoking by gender in the country.
Reviewer 4 Report
Comments and Suggestions for Authors
This pilot study evaluated a novel, inclusive national lung cancer screening program in Lithuania that invited individuals aged 50–70 regardless of smoking history to undergo low-dose chest CT (LDCT). Of 1,920 people invited, 1,014 participated (76.1% participation rate), with 1.4% showing Lung-RADS category 4 nodules and one confirmed case of squamous cell carcinoma. The screening also uncovered significant incidental findings in 30.1% of participants, including cardiovascular and pulmonary conditions, prompting specialist referrals in 17.6% of cases. The study demonstrated the program’s feasibility, high acceptability, and organizational efficiency, aided by personal telephone invitations and AI-supported CT interpretation. These findings support broader national implementation and highlight the potential of inclusive LDCT screening to detect both malignant and non-malignant conditions early, enhancing public health outcomes. However, several sections of the manuscript require substantial revision and clarification to enhance its scientific rigor and readability before it can be considered for publication.
1.Abstract: Could the authors clarify in the abstract how many cancers were actually detected, not just Lung-RADS 4 nodules?
2.Introduction: While the background is thorough, could the authors further elaborate on why prior LDCT programs failed in Lithuania or in comparable regions?
3.Methods: Why was the age range of 50–70 selected, and could this potentially exclude younger at-risk individuals? In addition, Were participants informed about potential radiation risks, and how was this addressed in the consent process?
4.Results:
4.1 Given the 76.1% participation rate, were there any notable demographic or behavioral differences between participants and non-participants?
4.2 The high rate of incidental findings (30.1%) is significant—how were these triaged, and was overdiagnosis a concern?
5.Discussion:
5.1 The inclusion of non-smokers is ethically progressive—can the authors discuss the implications of this model for public health policy in other countries?
5.2 How do the results compare with large trials like NLST or NELSON in terms of cancer detection rates and incidental findings?
6.Conclusion: Will longitudinal follow-up be conducted to determine actual lung cancer incidence and survival outcomes in this cohort?
7.Could the distribution map of CT scanners (Figure 1) include population density or target population overlays for better context?
Author Response
Thank you for your detailed review of our manuscript and valuable suggestions. We have modified the manuscript based on your suggestions. The following is the point-to-point reply to the comments.
This pilot study evaluated a novel, inclusive national lung cancer screening program in Lithuania that invited individuals aged 50–70 regardless of smoking history to undergo low-dose chest CT (LDCT). Of 1,920 people invited, 1,014 participated (76.1% participation rate), with 1.4% showing Lung-RADS category 4 nodules and one confirmed case of squamous cell carcinoma. The screening also uncovered significant incidental findings in 30.1% of participants, including cardiovascular and pulmonary conditions, prompting specialist referrals in 17.6% of cases. The study demonstrated the program’s feasibility, high acceptability, and organizational efficiency, aided by personal telephone invitations and AI-supported CT interpretation. These findings support broader national implementation and highlight the potential of inclusive LDCT screening to detect both malignant and non-malignant conditions early, enhancing public health outcomes. However, several sections of the manuscript require substantial revision and clarification to enhance its scientific rigor and readability before it can be considered for publication.
RESPONSE: Thank you very much for your positive comment.
1.Abstract: Could the authors clarify in the abstract how many cancers were actually detected, not just Lung-RADS 4 nodules?
RESPONSE: Thank you for this question. Information on the detection of lung cancer is given in the summary (Lines 49-50 in the original manuscript).
2.Introduction: While the background is thorough, could the authors further elaborate on why prior LDCT programs failed in Lithuania or in comparable regions?
RESPONSE: You raise a very valid point. We understand why you're asking, but we cannot provide an analysis of failures, because such programs haven't existed in this region. Lung cancer screening has not been carried out in Lithuania before. National lung cancer screening is also not available in the neighbouring Baltic States – Latvia and Estonia. In neighboring Poland, it is just starting.
3.Methods: Why was the age range of 50–70 selected, and could this potentially exclude younger at-risk individuals? In addition, Were participants informed about potential radiation risks, and how was this addressed in the consent process?
RESPONSE: Thank you for this remark. The population of lung cancer patients primarily consists of individuals aged 50 to 70. It is a political decision of the authorities to launch a programme specifically for this age group. However, the age groups are likely to be extended in the future.
4.Results:
4.1 Given the 76.1% participation rate, were there any notable demographic or behavioral differences between participants and non-participants?
RESPONSE: Thank you for bringing up this comment. Unfortunately, we can only make assumptions. The call for the pilot study was conducted by phone. Most of those who declined to participate in the study were not interviewed in detail because they chose not to do so. However, we have included the available information relevant to your question in the manuscript.
4.2 The high rate of incidental findings (30.1%) is significant—how were these triaged, and was overdiagnosis a concern?
RESPONSE: Thank you very much for your question. When planning the program, we expected to find new significant co-foundations. Indeed, this is one of the two main objectives of our program. However, we did not anticipate such a high proportion of them. In each case, the clinical significance, the need to treat the patient, correct the patient's lifestyle, or refer the patient for specialist advice was determined individually by the family physician (Lines 292-295 in the original manuscript). The data are presented in Table 6 (formerly Table 7).
5.Discussion:
5.1 The inclusion of non-smokers is ethically progressive—can the authors discuss the implications of this model for public health policy in other countries?
RESPONSE: This is an excellent suggestion – thank you very much. We have added a few sentences about this in the conclusion section of the manuscript.
5.2 How do the results compare with large trials like NLST or NELSON in terms of cancer detection rates and incidental findings?
RESPONSE:
Comparing the data directly presents challenges due to the significant differences in study design and duration. For instance, the NLST study involved subjects who were followed for 5 to 7 years and underwent multiple rounds of low-dose computed tomography (LDCT). Notably, this study did not focus on actively screening for co-morbidities. Similarly, the NELSON study followed subjects for a minimum of 10 years, with participants also undergoing several rounds of LDCT, and it too did not aim to actively screen for co-morbidities.
6.Conclusion: Will longitudinal follow-up be conducted to determine actual lung cancer incidence and survival outcomes in this cohort?
RESPONSE: Thank you for the important remark. Yes, we are indeed planning to follow up on these subjects.
7.Could the distribution map of CT scanners (Figure 1) include population density or target population overlays for better context?
RESPONSE: Thank you for this remark. We have added the requested information to Figure 1.
Reviewer 5 Report
Comments and Suggestions for Authors
This is a report of the Lithuanian lung cancer screening program, which screened participants regardless of their smoking history. The authors claimed that they have demonstrated that their screening model is feasible, well-organized, and clinically valuable.
My comments:
- This manuscript is too long with very detailed description on every step they have done. Actually similar LDCT screening procedures were done world-wide, except all other screening programs would have more strict selection criteria, which is mainly to reduce the cost. According to the design of this pilot project, its cost (including the manpower) would be very high, but the authors did not discuss on this issue at all.
- The authors described that they have found 12 persons with lung-RADS 4A nodules, which included 8 nonsmokers and 4 smokers, and considered as an important result. But the authors did not show the final pathology diagnoses of these lung-RADS 4A nodules. They should give this data.
- The discussion is very long, redundant and difficult to read.
- The supplementary figures are not in English.
The English is not bad, but the manuscript is too long to read.
Author Response
This is a report of the Lithuanian lung cancer screening program, which screened participants regardless of their smoking history. The authors claimed that they have demonstrated that their screening model is feasible, well-organized, and clinically valuable.
RESPONSE: Thank you for reviewing our manuscript and your point. We appreciate your summary of our work.
My comments:
- This manuscript is too long with very detailed description on every step they have done. Actually similar LDCT screening procedures were done world-wide, except all other screening programs would have more strict selection criteria, which is mainly to reduce the cost. According to the design of this pilot project, its cost (including the manpower) would be very high, but the authors did not discuss on this issue at all.
RESPONSE: Thank you for your comment. Our programme concept is fundamentally different from those in some other countries. The three main differences are that non-smokers are also investigated, secondly that significant concomitant findings are actively investigated, and thirdly that it is carried out every three years (this is important in terms of subjects' radiation exposure and willingness to participate in repeated rounds) rather than every year.
The programme is rolled out in phases in a particular country. The first stage is to test the idea and the feasibility of the programme as a process. The second phase, which builds on the results of the first phase, is the financial evaluation. It is foreseeable, and planning has already started. This is indicated in the manuscript (original manuscript lines 388-390).
- The authors described that they have found 12 persons with lung-RADS 4A nodules, which included 8 nonsmokers and 4 smokers, and considered as an important result. But the authors did not show the final pathology diagnoses of these lung-RADS 4A nodules. They should give this data.
RESPONSE: Thank you for the remark. Early lung cancer screening is based on the detection of small nodules, their classification according to Lung-RADS, and standardised follow-up according to this classification for the vast majority of nodules. That's how it was done in our study. Most nodules will never have any histology. Because they will not grow. Only large nodules with signs of malignancy or growing foci are biopsied. But it may take a few more years for this to happen. Now, all the subjects in whom category 4 nodules have been found are actively and carefully followed up.
- The discussion is very long, redundant and difficult to read.
RESPONSE: Thank you for your point. Our study covers various aspects of lung cancer and its significant comorbid findings. In addition, it covers important issues related to the implementation of the programme as a process. Given that the concept of the programme is different from those existing in some other countries, we believe that the results should be presented and discussed in detail. As early diagnosis of lung cancer is still an unsolved problem, we expect that our model and its results (based on published data) will be thoroughly analysed by other researchers and health policy makers.
- The supplementary figures are not in English.
RESPONSE: Thank you for the remark. The additional figures are intended to illustrate the possibility of real-time monitoring of the lung cancer programme with all the necessary information. The legends of the figures describe in detail what is reflected in them. The information contained in the figures is not discussed in the manuscript as it is momentary for the specific study date. The point of the demonstration of the figures is to show the importance of an integrated tool to allow real-time monitoring of the process.
Round 2
Reviewer 4 Report
Comments and Suggestions for Authors Accept in the current version.Author Response
Thank you very much for your kind comment and positive rating!
Reviewer 5 Report
Comments and Suggestions for Authors
The authors did not make revision according to my comments. They still insist to write a very very long and detailed manuscript. They did not answer my questions about the cost-effectiveness. They said they have identified many other diseases, but they did not demonstrate or discuss on the impact or importance of those findings to the participants.
Author Response
The authors did not make revision according to my comments. They still insist to write a very very long and detailed manuscript. They did not answer my questions about the cost-effectiveness. They said they have identified many other diseases, but they did not demonstrate or discuss on the impact or importance of those findings to the participants.
RESPONSE: Thank you for your views. The concern you mentioned is thoroughly addressed in the first version of the revised manuscript, specifically from Lines 448 to 466.
Manuscript length — We intentionally retain detailed descriptions of the entire workflow, LDCT acquisition protocol, and management algorithm so that other health authorities can replicate or adapt the pilot without ambiguity.
Cost-effectiveness — The present manuscript reports only the clinical feasibility component of the pilot. A full health-economic evaluation will be conducted once adequate follow-up and resource-use data have accrued.
Impact of incidental findings — Participants are contacted by their family physician or team member to discuss the LDCT report. Follow-up timing is set according to the Lung-RADS category. Significant incidental findings trigger referral to the relevant specialty, and suspicious lesions are referred via a fast-track pathway. When smokers in the program meet with their family physician to discuss findings related to pulmonary issues or other significant health concerns, they are advised to quit smoking. These steps demonstrate the clinical importance of non-cancer findings and clarify how benefits extend beyond early lung-cancer detection.